# The sociomotor role and its meaning in traditional sporting games: Socio-affective rejection in question

Zhaïra Ben Chaâbane[1], Carlos Mallén-Lacambra[2], Aaron Rillo-Albert[2], Cristòfol Salas-Santandreu[2], Felipe Menezes-Fagundes[2], Verónica Alcaraz-Muñoz[3], Miguel Pic[4], Rosa Rodríguez-Arregi[2], Pere Lavega-Burgués[2]*

1 Motor Action Research Group (GIAM), Institut Supérieur du Sport et de l'Éducation Physique (ISSEP), Université de La Manouba, Tunis, Tunisia, 2 Motor Action Research Group (GIAM), INDEST, Institut Nacional d'Educació Física de Catalunya (INEFC), Universitat de Lleida (UdL), Lleida, Spain, 3 Motor Action Research Group (GIAM), UCAM Universidad Católica de Murcia, Murcia, Spain, 4 Motor Action Research Group (GIAM), Universidad de Valladolid, Campus Soria, Spain

* plavega@gencat.cat

## Abstract

### Purpose

This study analyses the sociomotor dynamics of the traditional sporting game (TSG) Bear, Guardian and Hunters from the perspective of motor praxeology, exploring how players adjust their strategies according to the internal logic of the game and their socio-affective status.

### Method

This is an observational study involving 10 university students, recording a total of 1,810 observational sequences during the game's development. Variables such as type of striking and distance from the Bear were assessed, along with sociometric indices of rejection and their influence on motor decisions.

### Results

The phases of the game condition players' behaviour. In the initial phase, motor aggressiveness predominates; in the intermediate phase, a defensive strategy is adopted; and in the final phase, extreme prudence prevails. Players with higher levels of social rejection exhibit more aggressive behaviour, and the Hunters tend to distance themselves more from the Bear when the latter is marginalised.

### Conclusion

This TSG provides a valuable educational setting for developing socio-affective relationships, managing motor aggressiveness, and fostering interpersonal relationships.

**Data availability statement:** The dataset has been deposited in a public repository, and can be accessed here: https://doi.org/10.5281/zenodo.17651091.

**Funding:** This research was funded by the National Institute of Physical Education of Catalonia (INEFC; code: DOGC No. 8568–22.12.2021, Resolution 21_03_2022), University of Lleida (UdL), through the project "OPPORTUNITY. Fostering social inclusion and gender equality in formal and non-formal educational contexts through applying traditional sports and games", co-funded by the Erasmus+ Programme of the European Union, project code: 622100-EPP-1-2020-1-ES-SPO-SCP. The funders had no role in study design, data collection and analysis, decision to publish, or preparation of the manuscript.

**Competing interests:** NO The authors have declared that no competing interests exist.

# Introduction

The development of socio-emotional competences and education in values such as coexistence, respect and empathy are essential in modern Physical Education. According to UNESCO [1] and the Kazan Action Plan [2], sport and physical activity not only benefit health, but also serve as key tools for social and emotional development. The 2030 Agenda [3] links Physical Education to Sustainable Development Goal 3 (SDG 3) (Health and well-being), SDG 4 (Quality education), SDG 5 (Gender equality) and SDG 16 (Peace and justice), promoting inclusion and educational equity. Within this framework, the Fit for Life programme [4] highlights the role of sport as a catalyst for social cohesion, developing essential life skills for living together in society, especially in schools. In this line, recent research highlights how motor games, as dynamic and participatory activities, can serve as powerful pedagogical tools to address key issues such as health, inclusion, environmental awareness, and social equity. By integrating physical activity with broader educational goals, motor play fosters not only physical and emotional development but also a commitment to sustainability and global well-being [5].

## Traditional sporting games: A valuable pedagogical resource for educating interpersonal relationships

Traditional Sporting Games (TSG) are a pedagogical resource that fosters social interaction through motor relationships. Previous research has shown that educational interventions with TSGs can enhance group dynamics and strengthen interpersonal relationships among students [6]. At the same time, other studies emphasize that pre-existing social configurations shape the motor interactions that take place during the game [7]. Together, these findings suggest that TSGs operate as a genuine relational laboratory, providing meaningful opportunities to foster students' education in peaceful coexistence.

From the perspective of motor praxeology, their internal logic defines the internal organisation of the game, determining the relationship with other players, space, time and equipment [8]. Each TSG possesses a unique internal logic, shaped by agreed-upon rules, functioning as a social laboratory where players experience different forms of interaction. Some encourage competition with winners and losers [9], while others promote the exchange of roles between cooperation and opposition [10]. An example is Bear, Guardian and Hunters, whose internal logic generates a continuous cycle of role changes in play, enriching the players' socio-affective dynamics.

## Educating the temporal framework in the traditional sporting game: "Bear, Guardian and Hunters"

The TSG Bear, Guardian and Hunters has a cyclical temporal structure, based on a continuous flow of roles without a predetermined outcome [11]. Players assume three successive roles: the Bear, tied by a rope to the Guardian, is passive and cannot defend themselves; the Hunters attempt to strike them with a handkerchief, while the Guardian tries to stop them by also striking with a handkerchief. Each impact

produces a role change, configuring a dynamic sociomotor network. This cyclical scheme regulates motor interaction and immerses players in a liminal time [12], where temporal perception varies depending on the role performed, shaping a flexible and unpredictable strategy.

### The regulation of motor behaviour: confrontation and guarding distances in the game "Bear, Guardian and Hunters"

To understand how players regulate motor aggressiveness and adjust their strategies, it is essential to analyse their relationship with the play space [13]. Proxemics allows us to examine how physical distances affect interactions [8], an approach applied by Parlebas [8] in his theory of motor action to study the management of distances in sociomotor dynamics.

In Bear, Guardian and Hunters, the distance of motor confrontation varies according to the strategy of the Hunters: when striking the Bear, they reduce the distance to an intimate level; when avoiding the Guardian, they adopt a personal distance. The guarding distance [14] influences the intensity of the confrontation. Hunters alternate between long distances to assess risk and short distances to attack, balancing motor aggressiveness with an effective defensive strategy.

### Control of motor aggressiveness and power relations in traditional sporting games

In Bear, Guardian and Hunters, the control of motor aggressiveness is key in sociomotor interaction. According to Elias and Dunning [15], sport regulates aggressiveness through rules that transform confrontation into a structured social ritual. Motor aggressiveness, governed within the ludic framework, is distinct from illicit violence [16]; Thus, Hunters may strike the Bear with a handkerchief within permitted limits, while any harmful action is sanctioned.

For Dugas [17] and Pfister [18], motor aggressiveness has educational value, channelling energy and fostering emotional self-control. Hunters, as agents of opposition, regulate the intensity of the strike, expressing their dominance over the Bear [19]. This regulation is an act of respect and empathy [20], aligning with Elias and Dunning [15] civilising process, in which dominance and sensitivity coexist in a complex socio-emotional experience.

In this TSG, the regulation of motor aggressiveness also follows implicit norms of coexistence [10]. According to Goffman [21], respect and dignity towards the opponent are essential in social interaction rituals. At an emotional level, players must balance their energy, maximising intensity without disrupting the harmony of the game [22]. In this line, previous studies on this game reported significant differences in the way men and women executed hitting actions, influenced by prevailing gender norms [6].

From the perspective of motor praxeology, Parlebas [8,23] distinguishes between motor behaviour (observable in displacements and strikes) and motor conduct, which integrates emotional and social meanings. In Bear, Guardian and Hunters, motor behaviour can be externally analysed, but its socio-affective impact requires a sociometric approach. This analysis shows that motor strategies depend on socio-affective status [6], providing an internal insight into the meaning of ludic action.

From the perspective of a Physical Education teacher, students build networks of role exchange and regulate motor aggressiveness. These external actions, defined as motor behaviours [8], are observable in displacements and strikes. However, when analysing their personal significance, the concept of motor conduct emerges, integrating physical, cognitive, emotional and social dimensions [23].

In Bear, Guardian and Hunters, motor conduct goes beyond movement execution; it involves interpreting the role, managing relationships and assigning emotional value to each action within the normative framework. To explore this dimension further, the sociometric survey offers a way to analyse the players' socio-affective status within the group and the possibility of comparing it with the motor strategies adopted, thus providing an internal perspective on the meaning of ludic performance.

**Analysing socio-affective relationships: A way of interpreting the meaning of players' motor behaviours**

The analysis of interpersonal relationships makes it possible to identify preferences and rejections within a group [24]. Sociometry, developed by Moreno [25], is a key tool in Physical Education for evaluating socio-affective relationships and guiding the pedagogy of motor conducts [8].

This study examines the influence of socio-affective rejection on motor aggression in the game Bear and Guardian. More specifically, it explores whether the level of socio-affective rejection affects the players' level of motor aggression: is motor aggression determined by the internal logic of the game and impacted by socio-affective factors? Do the Hunters, depending on their own level of socio-affective rejection and that of the Bear, modify their motor aggression? Are the more rejected Hunters more aggressive? Do they move closer to or further away from their "prey"? Do they attack more frequently or with greater intensity? Are they more aggressive towards the rejected Bear?

These questions explore the impact of social relationships on motor dynamics.

Previous studies have analysed the effect of TSGs on interpersonal relationships [26–28], but none has examined how social position influences motor behaviours [29,30]. Knowing students' sociometric position helps teachers design pedagogical strategies that take social rejection into account. More than an analytical tool, sociometry enables the assessment of the social climate and supports a dynamic, inclusive and collaborative pedagogy.

In light of this theoretical framework, the main aim of this study was to analyse the relationship between the sociomotor dynamics of the TSG Bear, Guardian and Hunters and the negative socio-affective status of the players. This aim was developed through two specific objectives:

a) To identify the motor behaviour patterns of the Hunters through observation during the three phases of the game (initial, intermediate, final), considering the duration of the sequence (relation to time), the distance (relation to space), and the type of striking (relation to equipment and to others).

b) To analyse the relationship between the Hunters' motor decisions towards the Bear and the players' socio-affective rejection.

These objectives are framed within the analysis of the game's internal logic and its capacity to generate a structure of sociomotor relationships in a controlled environment, contributing to a deeper understanding of the interrelation between a game's motor dynamics and interpersonal relationships in the field of Physical Education.

## Method

### Design

The application of an observational methodology should be regarded, in itself, as a mixed methods approach [31]. The present study applied a Type III design, inspired by the N/P/M observational methodology [32]. Accordingly, the Nomothetic (N) nature was justified by the interest in investigating collective units in TSGs. The design was considered Punctual (P) because only one recording was made. Moreover, linking Parlebas's [33] conceptualisation of roles and sub-roles with their corresponding criteria and categories allowed the study to be classified as Multidimensional (M).

### Participants

The study involved 10 students (5 women and 5 men; $M_{age} = 20.6$ years, $SD = 0.8$) from the double degree in Physiotherapy and Physical Activity and Sport, and from the degree in Physical Activity and Sport Sciences at the National Institute of Physical Education of Catalonia (INEFC), University of Lleida, Catalonia, Spain. A total of 1,810 observational sequences were recorded during the game, resulting in a high-density dataset that allowed for a detailed and robust analysis of motor behavior. The recruitment period for this study began on 13 February 2023 and ended on 8 May 2023. All participants provided written informed consent prior to participation in the study. The procedures adhered to the

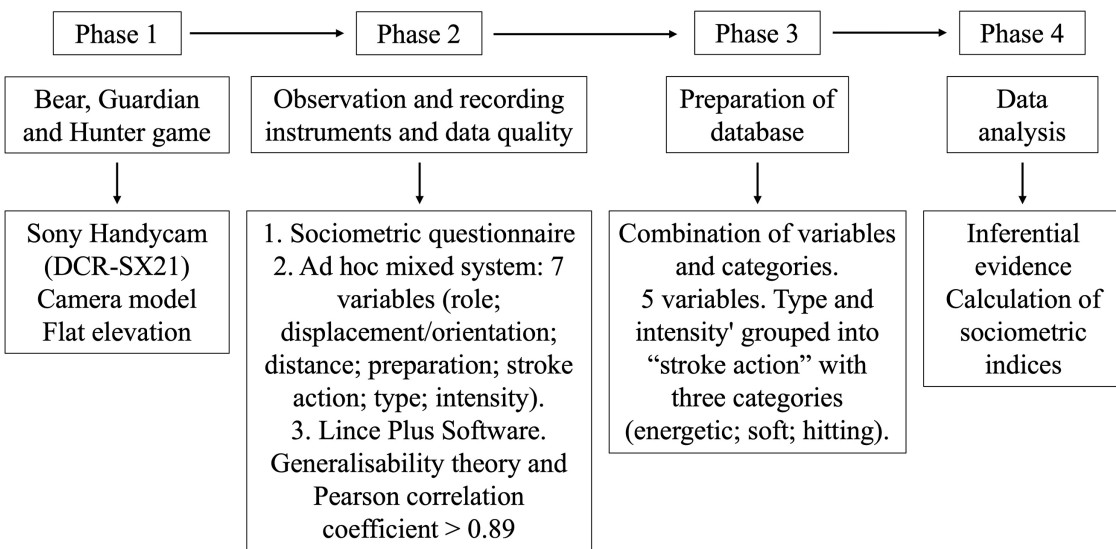

**Fig 1. The 4 phases followed in the methodological strategies of this research (Adapted from Muñoz-Arroyave et al. [34].**

Declaration of Helsinki and the ethical standards in sports science [34]. Ethical approval was obtained from the Ethics Committee for Clinical Research of the Catalan Sports Council (reference: 09/CEIGGC/2020).

## Procedure and materials

The methodological part of the study was developed in four phases (see Fig 1).

**Phase 1. The "Bear – Guardian – Hunters" game.** The players were organised into a single group and played the TSG Bear, Guardian and Hunters. The match lasted 8 minutes, as this is the duration typically used by the GIAM (Motor Action Research Group) in previous studies on other traditional sporting games [26,27,35]. While the game could have been developed for a longer period, we considered the use of 8 minutes to be prudent in order to allow for comparisons with other groups, without interruptions. Moreover, the participants' engagement might also have been diminished if the game were excessively prolonged. The game was recorded using a Sony Handycam (model DCR-SX21) from an elevated position to facilitate subsequent analysis.

**Phase 2. Instruments and data quality.** Sociometric questionnaire. To assess the socio-affective relationships prior to the intervention, a sociometric questionnaire with unlimited, directed and weighted nominations was used, following the methodology of Avramidis et al. [36].

The questionnaire was administered in the presence of all members of the group. This collective format of the questionnaire has the advantage of bringing together all the people who are both subjects and objects of preferences or rejections [37]. To ensure privacy, participants completed the questionnaire individually and were informed that their answers would remain confidential, influencing the formation of groups for a final recreational, non-competitive activity. These strategies, recommended by Parlebas [37], allowed for the collection of authentic socio-affective networks [24], minimising instrumental responses.

For the purposes of our study, the analysis of the sociometric questionnaire data focused primarily on the response to the socio-affective rejection item: "With whom would you not like to share the final activity?"

Observation instrument and recording of observational data. The Motor Action Research Group (GIAM) designed an *ad hoc* mixed recording system with exhaustive and mutually exclusive categories [38]. Initially, roles and sub-roles were identified deductively based on the theoretical framework of Parlebas [8,39], and were later expanded inductively from observations, incorporating new categories such as the Transition role. After defining 21 variable categories (role,

handkerchief preparation, striking action and type, intensity, displacement/orientation, and distance), a detailed manual was produced for observers, describing each category and its degrees of freedom.

The motor behaviours during the game were recorded using the free LINCE PLUS software [40], which enables visualisation and coding of recorded footage.

To ensure data quality, four pairs of researchers from GIAM were selected according to their experience in systematic observation (a minimum of two years using the methodology). Following several training sessions, each pair independently recorded the behaviour of one participant over eight minutes of gameplay. Data quality was assessed using Generalizability Theory [41]. High variance was found in the Category facet [C] (99.207%) and no variance in the Observer facet [O], with a marginal interaction [C][O] (0.793%). In addition, Pearson correlation coefficients between observers exceeded 0.98, confirming the reliability of the data.

**Phase 3. Database preparation.** After data collection, categories with low frequencies or limited information were reclassified to optimise statistical analysis and interpretation. The category Multiple strikes was integrated into Striking, distinguishing between soft and vigorous according to intensity. Type of striking (prepared/unprepared) and handkerchief preparation were merged into Strike preparation, encompassing aggressive preparatory actions. Waiting and no displacement were unified under No movement. The role transition category was reclassified as Hunter, since it did not entail a genuine role change. This process consolidated the instrument into five variables (role, displacement/orientation, distance, preparation, and striking action), with a total of 15 categories.

**Phase 4. Data analysis.** In the present study, multiple statistical analyses were carried out in order to examine the relationships between the sociomotor variables observed in the TSG Bear, Guardian and Hunters. Inferential tests were used to evaluate associations between independent and dependent variables, as well as the calculation of sociometric indices that provided information on the social dynamics among players.

Independent variables. The analysis considered independent variables that reflect the structure of the game and the sociometric dynamic:

• Game phase: initial, intermediate, final and transitional.

• Sequence duration: short (1–3 seconds) or long (more than 3 seconds).

• Sociometric rejection: calculated using UCINET through closeness, categorising players into High Rejection and Low Rejection [42]. The classification was made using the midpoint of the empirical range of the observed scores (minimum = 9, maximum = 45). Participants with a score below 27 on proximity to rejection were classified in the High Rejection group, while those with a score above 27 were placed in the Low Rejection group. It should be noted that proximity to rejection is an inverse indicator, so lower values reflect higher levels of rejection.

This approach standardises the interpretation of rejection, allowing its impact on Hunters and Bears to be assessed in each game sequence.

To measure social rejection, the closeness index was used, indicating the degree of connection of a player within the group based on sociometric distances. A high closeness value in rejection reflects a high level of exclusion [43], providing key information about socio-affective dynamics and their impact on motor interaction.

Dependent variables. The dependent variables evaluated the motor behaviour of the players during the game. These included: • Sequence duration: short or long (relation to time).

• Distance between the Hunter and the Bear: classified as short, medium or long (relation to space).

• Use of the handkerchief: whether or not the handkerchief was prepared to increase firmness (relation to equipment).

• Type of striking: soft strikes and vigorous strikes (relation to others).

Statistical tests. The following statistical tests were used to examine the relationships between categorical variables:

a) **Likelihood Ratio Chi-Square Test ($\chi^2$):** This test was employed to determine the statistical significance of the relationships between independent and dependent variables by comparing observed and expected frequencies under the null hypothesis. $\chi^2$ values ($\chi^2\_LR$) are reported alongside their corresponding degrees of freedom ($df$), p-values, and *Cramér's V* index, which measures the strength of association. The displacement analysis revealed a significant association between the game phase and type of displacement ($\chi^2 = 52.920$, $p < .001$), although the strength of association was low (*Cramér's V* = .099).

b) **Adjusted Residual Analysis:** This was used to identify which cells in the contingency tables contributed most to the observed significant differences. Adjusted residuals indicate whether the observed frequencies in a category differ significantly from the expected ones. Adjusted standardised residuals that differed significantly from expected values were marked with one asterisk for *z-scores* between 1.96 and 2.57 ($p < .05$), two asterisks for scores between 2.58 and 3.29 ($p < .01$), and three asterisks for scores above 3.29 ($p < .001$), as recommended by APA 7.0 guidelines.

c) **Effect size:** The effect size of associations between categorical variables was assessed using *Cramér's V* index, following Cohen's (1988) criteria. The effect was considered negligible ($V < 0.10$), weak ($0.10 \leq V < 0.20$), moderate ($0.20 \leq V < 0.40$), relatively strong ($0.40 \leq V < 0.60$), and strong ($V \geq 0.60$).

d) **Contingency table integration:** Each contingency table analysed aspects of motor behaviour and sociometric relationships. In each subtable, the count, row percentages, significance level of adjusted residuals, $\chi^2$ and p-values from the likelihood ratio test (*LR*), and the strength of association via Cramér's V were calculated.

## Results

As shown in Tables 1, 2 and 3, the statistical analyses of this study produced three tables, which are presented below to facilitate the understanding of this section.

**Table 1. Relationship between the dependent variable hit type and each of the independent variables separately.**

| Variables | Categories | Total (N=1810) | No hit (n=1513) | Preparation (n=14) | Soft hit (n=206) | Energetic hit (n=77) | $\chi^2_{LR}$ | $p_{LR}$ | Cramer's V |
|---|---|---|---|---|---|---|---|---|---|
| Sequence | Initial | 283 (15.6%) | 211 (74.6%) | 2 (0.7%) | 47 (16.6%) | 23 (8.1%) | 54.676 | <.001*** | .099 (n) |
| | Intermediate | 450 (24.9%) | 358 (79.6%) | 1 (0.2%) | 67 (14.9%) | 24 (5.3%) | | | |
| | Final | 346 (19.1%) | 288 (83.2%) | 5 (1.4%) | 36 (10.4%) | 17 (4.9%) | | | |
| | Transition | 731 (40.4%) | 656 (89.7%) | 6 (0.8%) | 56 (7.7%) | 13 (1.8%) | | | |
| Duration | Brief (1–3 s) | 1544 (85.3%) | 1248 (80.8%) | 13 (0.8%) | 206 (13.3%) | 77 (5.0%) | 99.856 | <.001*** | .181 (w) |
| | Extensive (> 3 s) | 266 (14.7%) | 265 (99.6%) | 1 (0.4%) | 0 (0.0%) | 0 (0.0%) | | | |
| Level of rejection of the Hunter | Low | 835 (46.1%) | 719 (86.1%) | 2 (0.2%) | 103 (12.3%) | 11 (1.3%) | 44.391 | <.001*** | .146 (w) |
| | High | 975 (53.9%) | 794 (81.4%) | 12 (1.2%) | 103 (10.6%) | 66 (6.8%) | | | |
| Level of rejection of the Bear | Low | 497 (27.5%) | 411 (82.7%) | 7 (1.4%) | 56 (11.3%) | 23 (4.6%) | 3.476 | .324 | .046 (n) |
| | High | 1313 (72.5%) | 1102 (83.9%) | 7 (0.5%) | 150 (11.4%) | 54 (4.1%) | | | |

*Note.* This table presents 11 contingency subtables (e.g., subtable relates hit type with sequence) with the following statistics: count, row percentage, significance level for positive adjusted standardized residuals (z-scores), $\chi^2$ and p-value for likelihood-ratio tests (*LR*), and *Cramer's V* measure of association. The strength of the association is categorized as follows: n = negligible association (.00 and under .10); w = weak association (.10 and under .20); m = moderate association (.20 and under .40); rs = relatively strong association (.40 and under .60); s = strong association (.60 and under .80); vs = very strong association (.80 and under 1.00). *$p < .05$. **$p < .01$. ***$p < .001$.

**Table 2. Relationship between the dependent variable displacement and each of the independent variables separately.**

| Variables | Categories | Total (N=1810) | Back (n=382) | No move (n=702) | Side (n=191) | Fordward (n=535) | $\chi^2_{LR}$ | $p_{LR}$ | Cramer's V |
|---|---|---|---|---|---|---|---|---|---|
| Sequence | Initial | 283 (15.6%) | 65 (23.0%) | 89 (31.4%) | 38 (13.4%) | 91 (32.6%) | 52.920 | <.001*** | .099 (n) |
| | Intermediate | 450 (24.9%) | 102 (22.7%) | 140 (31.1%) | 59 (13.1%) | 149 (33.1%) | | | |
| | Final | 346 (19.1%) | 80 (23.1%) | 118 (34.1%) | 36 (10.4%) | 112 (32.4%) | | | |
| | Transition | 731 (40.4%) | 135 (18.5%) | 355 (48.6%) | 58 (7.9%) | 183 (25.0%) | | | |
| Duration | Brief (1–3 s) | 1544 (85.3%) | 379 (24.5%) | 444 (28.8%) | 188 (12.2%) | 533 (34.5%) | 495.410 | <.001*** | .496 (rs) |
| | Extensive (> 3 s) | 266 (14.7%) | 3 (1.1%) | 258 (97.0%) | 3 (1.1%) | 2 (0.8%) | | | |
| Level of rejection of the Hunter | Low | 835 (46.1%) | 182 (21.8%) | 333 (39.9%) | 83 (9.9%) | 237 (28.4%) | 2.108 | .551 | .034 (n) |
| | High | 975 (53.9%) | 200 (20.5%) | 369 (37.8%) | 108 (11.1%) | 298 (30.6%) | | | |
| Level of rejection of the Bear | Low | 497 (27.5%) | 105 (21.1%) | 202 (40.6%) | 43 (8.7%) | 147 (29.6%) | 3.049 | .384 | .040 (n) |
| | High | 1313 (72.5%) | 277 (21.1%) | 500 (38.1%) | 148 (11.3%) | 388 (29.6%) | | | |

*Note*. This table presents 11 contingency subtables (e.g., one subtable relates displacement with duration) with the following statistics: count, row percentage, significance level for positive adjusted standardized residuals ($z$-scores), $\chi^2$ and p-value for likelihood-ratio tests (LR), and *Cramer's V* measure of association. The strength of the association is categorized as follows: n = negligible association (.00 and under .10); w = weak association (.10 and under .20); m = moderate association (.20 and under .40); rs = relatively strong association (.40 and under .60); s = strong association (.60 and under .80); vs = very strong association (.80 and under 1.00). *p < .05. **p < .01. ***p < .001.

**Table 3. Relationship between the dependent variable distance and each of the independent variables separately.**

| Variables | Categories | Total (N=1810) | Long (n=218) | Medium (n=782) | Short (n=810) | $\chi^2_{LR}$ | $p_{LR}$ | Cramer's V |
|---|---|---|---|---|---|---|---|---|
| Sequence | Initial | 283 (15.6%) | 39 (13.8%) | 100 (35.3%) | 144 (50.9%) | 31.530 | <.001*** | .093 (n) |
| | Intermediate | 450 (24.9%) | 62 (13.8%) | 167 (37.1%) | 221 (49.1%) | | | |
| | Final | 346 (19.1%) | 41 (11.8%) | 144 (41.6%) | 161 (46.5%) | | | |
| | Transition | 731 (40.4%) | 76 (10.4%) | 371 (50.8%) | 284 (38.9%) | | | |
| Duration | Brief (1–3 s) | 1544 (85.3%) | 202 (13.1%) | 604 (39.1%) | 738 (47.8%) | 71.775 | <.001*** | .199 (w) |
| | Extensive (> 3 s) | 266 (14.7%) | 16 (6.0%) | 178 (66.9%) | 72 (27.1%) | | | |
| Level of rejection of the Hunter | Low | 835 (46.1%) | 85 (10.2%) | 365 (43.7%) | 385 (46.1%) | 5.253 | .072 | .054 (n) |
| | High | 975 (53.9%) | 133 (13.6%) | 417 (42.8%) | 425 (43.6%) | | | |
| Level of rejection of the Bear | Low | 497 (27.5%) | 39 (7.8%) | 220 (44.3%) | 238 (47.9%) | 12.601 | .002** | .080 (n) |
| | High | 1313 (72.5%) | 179 (13.6%) | 562 (42.8%) | 572 (43.6%) | | | |

*Note*. This table presents 11 contingency subtables (e.g., one subtable relates distance with duration) with the following statistics: count, row percentage, significance level for positive adjusted standardized residuals ($z$-scores), $\chi^2$ and p-value for likelihood-ratio tests (LR), and *Cramer's V* measure of association. The strength of the association is categorized as follows: n = negligible association (.00 and under .10); w = weak association (.10 and under .20); m = moderate association (.20 and under .40); rs = relatively strong association (.40 and under .60); s = strong association (.60 and under .80); vs = very strong association (.80 and under 1.00). *p < .05. **p < .01. ***p < .001

## Sociomotor dynamics of the game

This section analyses the results of the observation of players' motor behaviour, without taking sociometric values into account, focusing on variables such as the duration of game sequences, the distance between the Hunter and the Bear, and the type of striking.

**Distance adopted by the Hunters in relation to the Bear and sequence duration.** The results indicate that the distance between the Hunters and the Bear significantly influences the duration of the sequences ($\chi^2 = 71.775$, $p < .001$), showing a moderate association (*Cramér's V* = 0.199). As shown in Table 3, when most Hunters choose to stay close to the Bear, the sequences are brief (1–3 seconds). Conversely, when fewer players approach, the

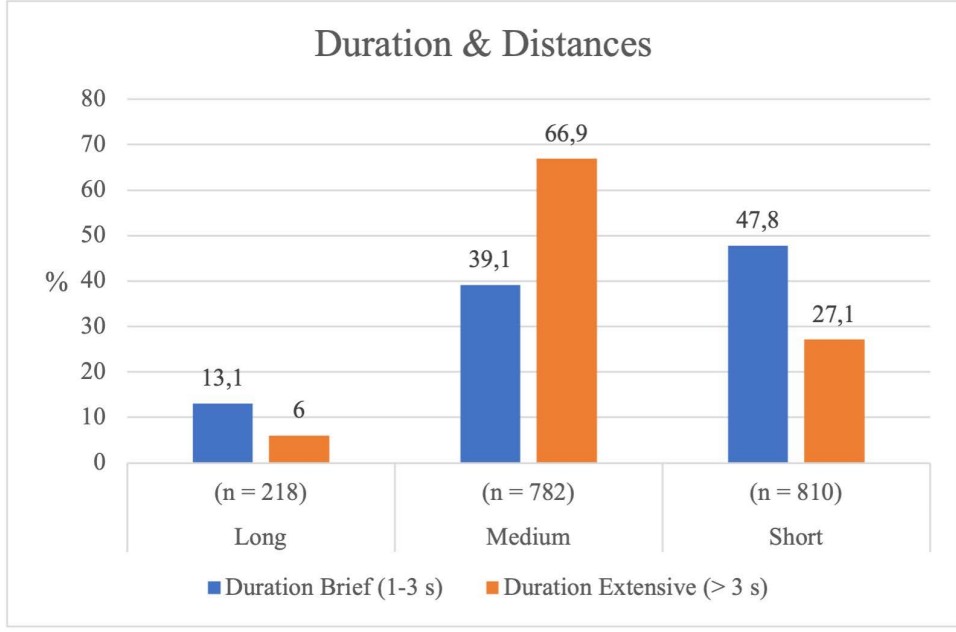

**Fig 2. Distance of the Hunter from the Bear according to the duration of the sequences.**

sequences are longer (>3 seconds) (see Fig 2). In general, the greater the distance, the longer the sequence. When adopting medium distances, Hunters tend to prolong the sequence (Table 3). Only a few remain far from the Bear, confirming that players prefer to move closer in order to advance, strike, and alternate roles—consistent with the game's internal logic. The level of risk is thus directly linked to the distance from the Bear. The observed motor behaviours reaffirm the rules of the game, illustrating how proximity and risk-taking jointly shape the temporal structure of each sequence.

**Distance adopted by the Hunters in relation to the Bear and game phases.** The analysis of the distance between the *Hunters* and the *Bear* across the different game phases revealed significant differences ($\chi^2 = 31.530$, $p < .001$), although the association was weak (*Cramér's V* = 0.093).

In the initial phase, most *Hunters* positioned themselves close to the *Bear*, indicating a greater propensity to take risks. In the intermediate phases, this proximity gradually decreased, reflecting a shift toward more cautious behaviour (see Table 3 and Fig 3).

Medium distances increased progressively from the initial to the final phase, suggesting the adoption of a more conservative strategy as the game advanced. Long distances, however, remained stable throughout all phases and were the least preferred by the *Hunters*, indicating that keeping far from the *Bear* was not a favoured option.

Despite the weak statistical association, these findings are consistent with the internal logic of the game, which encourages proximity to facilitate role alternation and dynamic interaction among players.

**Types of striking and sequence duration.** The type of striking had a significant effect on sequence duration ($\chi^2 = 99.856$, $p < .001$), showing a moderate association (*Cramér's V* = 0.181). In short sequences, *soft hits* occurred more frequently than *hard hits*, whereas in longer sequences only a negligible number of *soft hits* was observed (see Table 1 and Fig 4). With appropriate caution, it can be suggested that this pattern is consistent with the internal logic of the game, whose aim is to strike the *Bear* in order to trigger a role change. The low rate of successful strikes may support this interpretation, as it would help reduce passivity and maintain active engagement throughout the game.

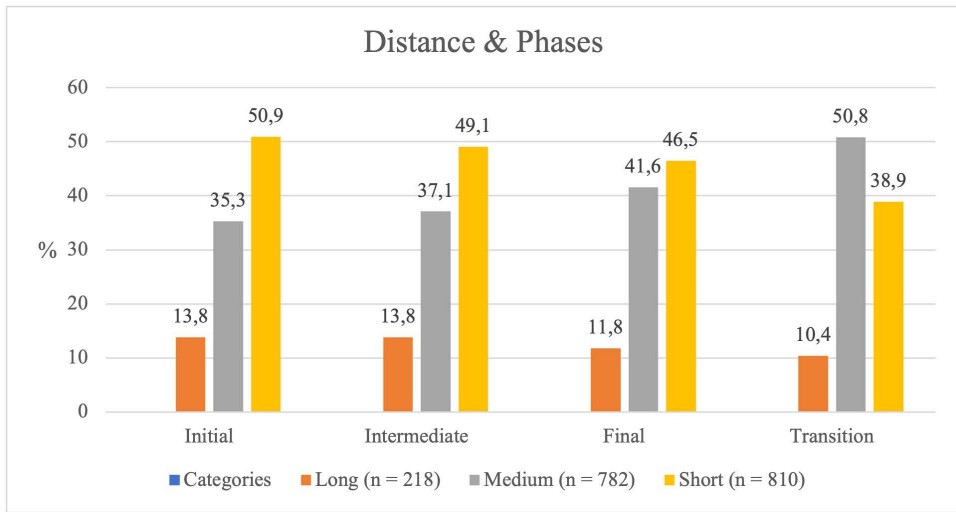

**Fig 3. Hunter's distance from the Bear depending on the game phase.**

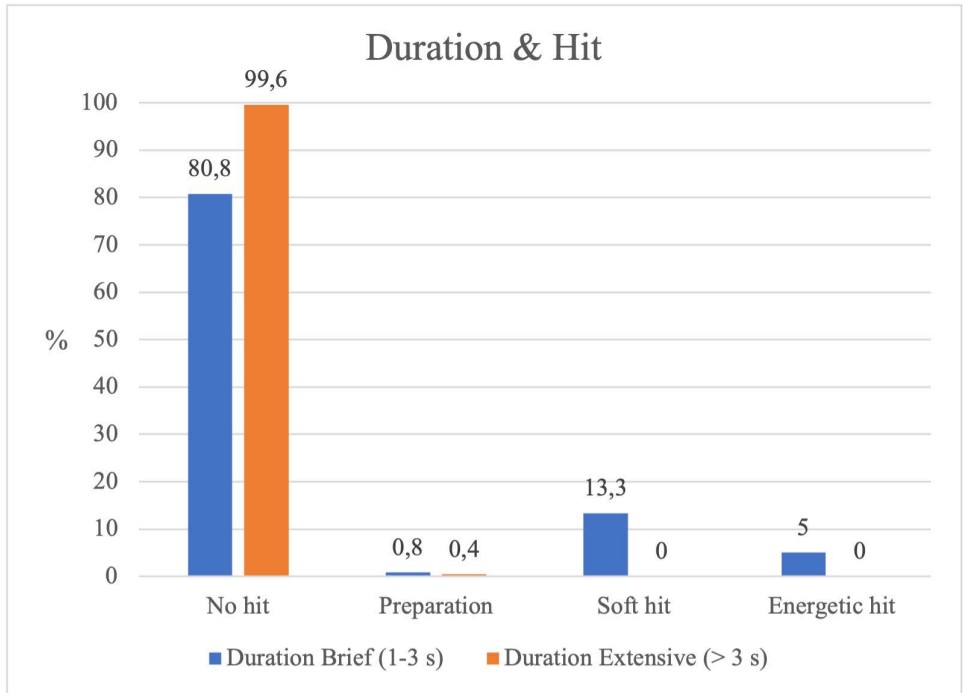

**Fig 4. Type of hit according to the duration of the sequence.**

The results reveal a significant relationship between the Hunters' striking actions and sequence duration. Initial motor aggressiveness shortens the sequences, while reduced involvement towards the end prolongs them.

**Type of striking and game phases.** The type of striking differed significantly across the game phases ($\chi^2 = 54.676$, $p < .001$), although the association was weak (*Cramér's V* = 0.099). During the initial phase, *Hunters* executed the highest

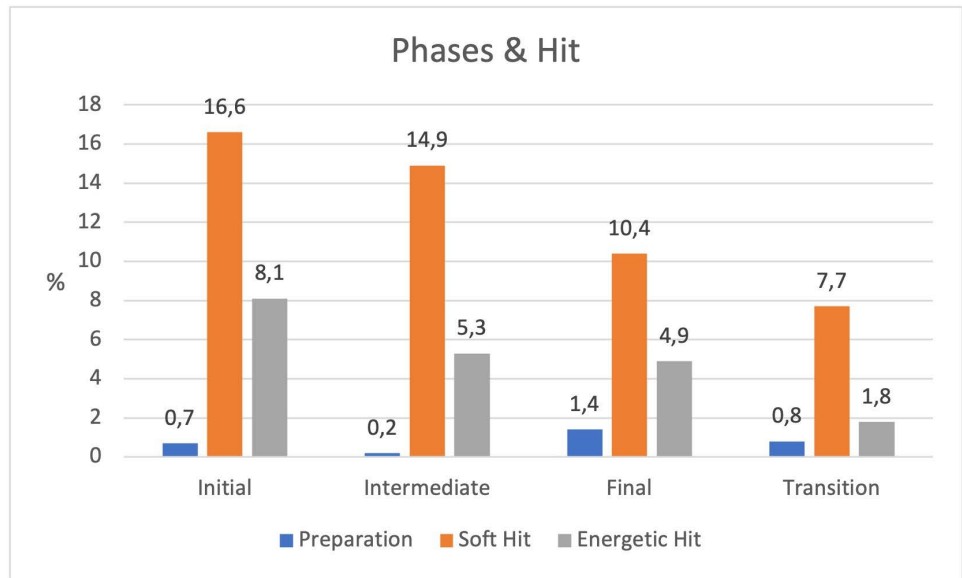

**Fig 5. Type of hit according to the duration of the game phase.**

number of strikes, with *soft hits* being far more frequent than *hard hits* (see Table 1 and Fig 5). As the game progressed, the frequency of striking actions decreased, indicating a lower tendency to attack in later phases. During transition moments—when the rules required the players to halt their actions—several *Hunters* nevertheless attempted to strike the *Bear* (see Table 1), taking advantage of its temporary vulnerability by delivering vigorous hits. This behaviour may suggest an attempt to complete a previously unsuccessful action, even at the cost of violating the rule. For these players, achieving the goal appears to have taken precedence over adhering to the game's constraints, revealing a tension between *strategic intent* and *regulatory compliance* that reflects the game's internal dynamics.

The data indicate that Hunters are more proactive at the beginning of the game, with greater frequency and intensity of striking. However, as the game progresses, they adopt a more conservative strategy, reducing both the intensity and frequency of their strikes. This may be due to the increased risk of being caught, or to the need to adapt to a game dynamic that encourages more careful and reflective decision-making.

### Influence of socio-affective relationships on the Hunters' motor decisions towards the Bear

A significant variability in rejection levels was identified, distinguishing between players with high and low rejection. These differences directly affected motor conduct, highlighting the impact of social rejection on the game's dynamics.

**Distance between Hunters and the Bear and sociometric indices.** *Relationship between the Bear's level of rejection and the distance maintained by the Hunters.* The Bear's level of rejection influenced the Hunters' distance ($\chi^2 = 12.601$, $p = .002$), with a small association (*Cramér's V* = 0.080).

When facing a Bear with high rejection (BRejHigh), Hunters were more likely to maintain a long distance than when interacting with a Bear with low rejection (BRejLow) (see Table 3). Moreover, Hunters tended to approach BRejLow Bears more closely than BRejHigh ones.

Given the moderate statistical association, it may be suggested that Hunters adopted a more conservative strategy when confronting a socially excluded Bear, maintaining greater distance throughout the game (see Fig 6).

*Relationship between the Hunter's level of rejection and the distance from the Bear.* The level of rejection of the Hunter did not have a significant impact on the distance maintained in relation to the Bear ($\chi^2 = 5.253$, $p = .072$; *Cramer's*

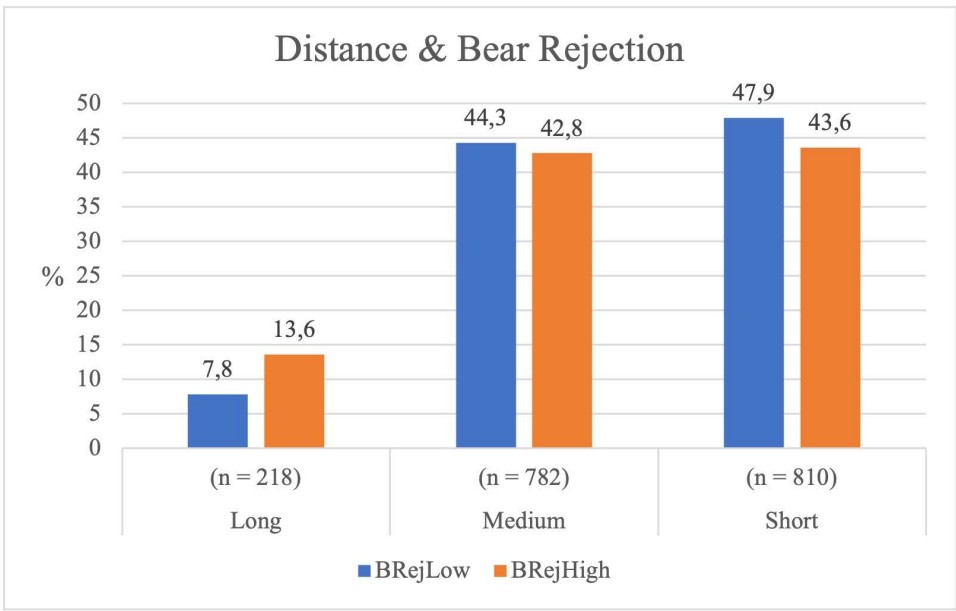

**Fig 6. Hunter's distance from the Bear according to the Bear's rejection level.**

$V = 0.054$). In terms of motor behaviour, Hunters with low rejection (HRejLow) maintained a short distance 46.1% of the time, whereas those with high rejection (HRejHigh) did so 43.6% of the time. Although less rejected Hunters tended to approach the Bear more often, the differences were not statistically significant (see Fig 7).

In summary, the statistical analysis revealed that rejection towards the Bear significantly influenced the Hunters' motor decisions: the more rejected the Bear, the greater the tendency to maintain distance. In contrast, the rejection of the Hunters did not have a significant impact on their own decision to approach or move away from the Bear.

**Striking actions performed by Hunters and sociometric indices.** *Relationship between the Hunter's level of Rejection and the Type of Striking.* The *Hunters'* level of socio-affective rejection significantly influenced the type of striking performed ($\chi^2 = 44.391$, $p < .001$), showing a moderate association (*Cramér's V = 0.146*). *Low-rejection Hunters* (*HRejLow*) refrained from hitting in the vast majority of cases. When they did strike, their actions were predominantly soft rather than vigorous. In contrast, *High-rejection Hunters* (*HRejHigh*) struck more frequently ($p < .001$) and with greater intensity (see Table 1 and Fig 8).

These results indicate that Hunters with higher socio-affective rejection exhibited more hostile motor behaviour towards the Bear compared to those who were less rejected.

*Relationship between the Bear's level of rejection and the type of striking by the Hunters.* The *Bear's* level of socio-affective rejection did not significantly influence the *Hunters'* type of striking ($\chi^2 = 3.476$, $p = .324$), showing a weak association (*Cramér's V = 0.046*). The *Hunters* displayed similar motor behaviour regardless of the *Bear's* rejection status.

The vast majority of *Hunters* refrained from striking the *Bear*, whether it was highly or weakly rejected, and the proportion of soft strikes remained almost identical across both conditions (see Table 1 and Fig 9).

Overall, the decision to strike or not to strike the *Bear* did not vary according to its level of socio-affective rejection, suggesting that this factor had little influence on the players' motor responses.

While highly rejected Hunters showed greater frequency and intensity of striking, the Bear's socio-affective status did not significantly affect the Hunters' offensive decisions.

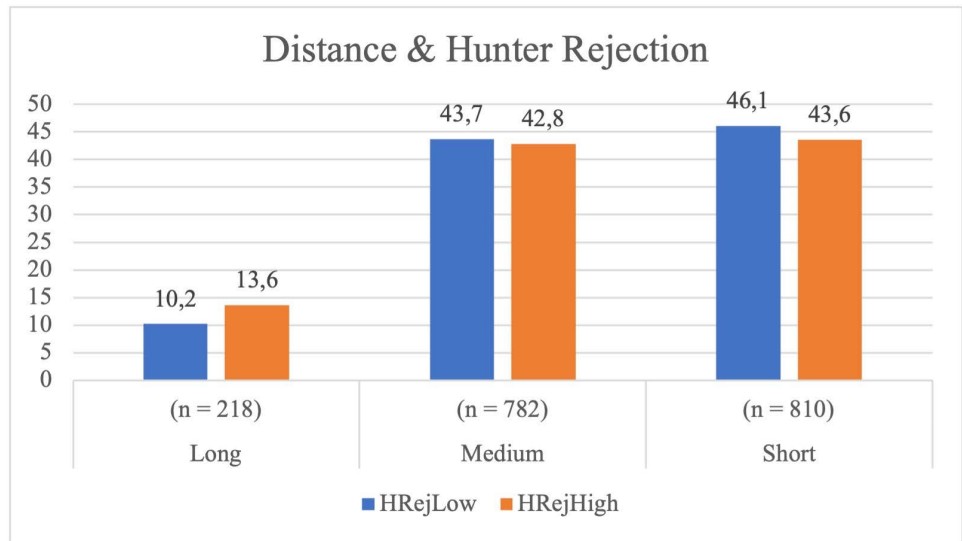

**Fig 7. Hunter's distance from the Bear according to the Hunter's rejection level.**

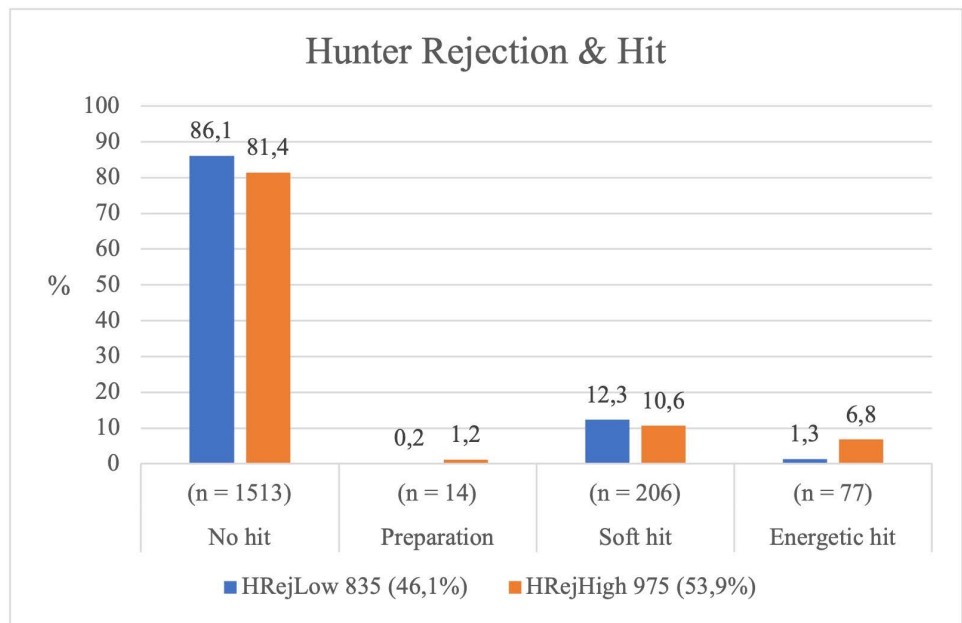

**Fig 8. Type of hit according to the Hunter's rejection level.**

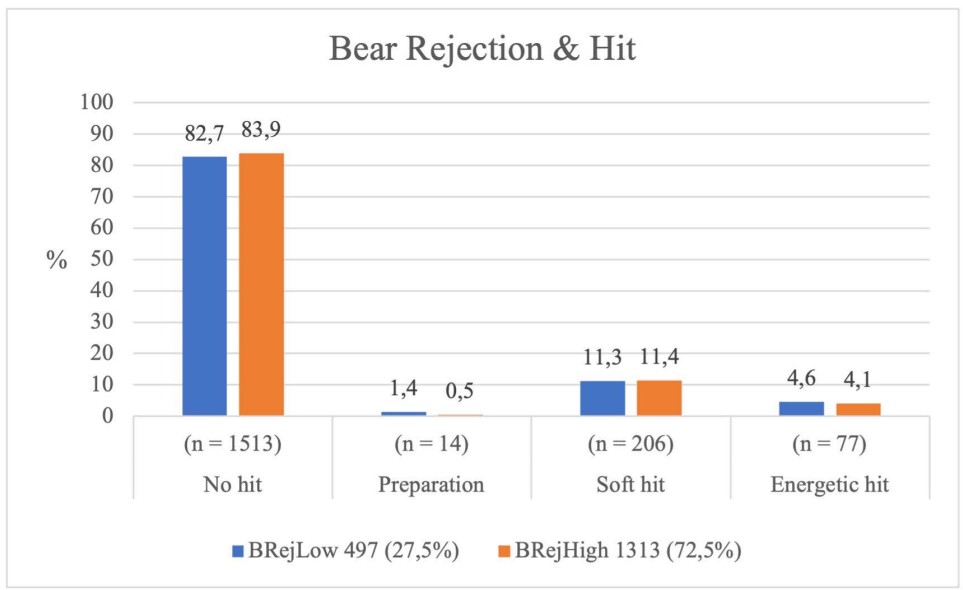

**Fig 9. Type of hit according to the Bear's rejection level.**

## Discussion

This study analysed the motor behaviour of the Hunters in the traditional sporting game (TSG) Bear, Guardian and Hunters. First, it examined how they adjusted their motor strategies to the internal logic of the game, adapting to its phases and the exchange of roles. In the second part, the relationship between these motor decisions and their socio-affective status within the group was explored, considering the impact of socio-affective rejection both of the Bear and of the Hunters.

### Distance between the Hunters and the Bear: a strategic choice between risk-taking and prudence

The results show that the majority of Hunters maintained a short distance from the Bear (~50%), compared to those who opted for a medium (35–42%) or long distance (12–14%). The group remained active, taking risks by positioning themselves within or near the Bear's circle, which increased the likelihood of being caught by the Guardian and undergoing a role change.

These risky behaviours involve invading the Bear's intimate space [44] and reducing the guard distance to zero [8]. The frontal and impulsive attack seeks to surprise the Guardian and maximise the Hunter's motor expectation of successfully striking the Bear.

In this research, maintaining a long distance was relatively infrequent, suggesting that prudence was not the preferred option among players. This finding indicates that the sociomotor dynamics of the game may influence players' strategic decisions. In the initial phase, the Hunters take advantage of the element of surprise, approaching the Bear boldly before the Guardian fully adapts to the defensive role. In the intermediate phase, as the Guardian's defence improves, the Hunters adopt a more cautious strategy, increasing the guard distance to 37.1%. This distancing reflects a more accurate assessment of risk while planning the next action.

Long sequences indicate that the Hunters have uncovered the "secret code" of the game: it is time for reflection. In these phases, they appear to take more time to optimise their strategies, favouring a time-related behaviour that enables

planning. From an educational perspective, this dynamic may hold significant pedagogical value, as it can foster motor intelligence grounded on reflection-in-action, within a ritual of respectful interpersonal relationships [22,45].

As the game progresses, the Hunters reduce their approach towards the Bear (initial phase = 50.9%; final phase = 46.5%) and increase medium distances (initial phase = 35.3%; final phase = 50.8%). Long distances decrease slightly (initial phase = 13.8%; final phase = 10.4%), without eliminating the risk of being caught by the Guardian. This willingness to take risks reflects the relationship between power, status, and decision-making in contexts of interaction [19].

## Game dynamics: Role changes and sociomotor intensity

During the game, 39 role changes were recorded, highlighting the intense sociomotor dynamic of the game. The data reveal a direct relationship between the Hunters' guard distance and the duration of the sequences. With a short distance, short sequences (~50%) prevail, whereas with a medium distance, long sequences increase (~70%). The interruption of the sequence depends on proximity to the Bear: risk-taking behaviours generate shorter sequences and provoke more role changes, intensifying emotional energy. This supports Collins' [22] theory of interaction rituals, according to which situations of high emotional intensity strengthen social bonds and game dynamics.

Short sequences were also observed at medium distances, suggesting that the Hunters, even in risky positions, managed to avoid being caught by the Guardian, thereby preventing role change and prolonging the game sequence. This pattern reveals a correspondence between the relationship with time and the relationship with space, activating the game's internal logic [23]. Motor strategies and decisions based on distance allow players to explore and manage risk within the structure of established roles and relationships.

## Types of strikes by the Hunters: Transition from motor aggressiveness to moderation

A global view of the game shows that the Hunters mostly use soft strikes with the handkerchief, while energetic strikes are less frequent. As the game progresses, both types of striking decrease, reflecting a reduction in the intensity and frequency of oppositional motor interactions. Observations suggest that the game unfolds with regular role changes in a relaxed environment, without an obsession with displaying dominance over the Bear [19]. The patterns observed suggest that the group may have internalised respectful behaviour, reconciling risk-taking with controlled expressions of motor aggressiveness in their interactions.

The results reveal a significant relationship between the Hunters' strikes, the game phases and the duration of the sequences. Throughout the game, moments without contact between Hunters and the Bear prevail, both in long and short sequences. Short sequences may be explained by the Guardian's capture of the Hunters, especially when they employ medium or short guard distances, increasing the risk of role change.

When soft or energetic strikes occur, sequences tend to be short, suggesting that the risk taken by the Hunters makes them more vulnerable to the Guardian's opposition. Risk, power, emotion and strategy form an interconnected phenomenon within the Hunters' motor strategies [11]. Observing their motor behaviours in relation to space (guard distances), time (sequence duration, phases), and strikes to the Bear confirms an intervention aligned with the internal logic of the game Bear, Guardian and Hunters.

This game is characterised by a frequent role-changing system, influenced by the duration of sequences and short opposition distances, which favours offensive and risky actions essential to its progression. However, as the game advances, players adjust their behaviour according to the game's internal logic, which governs their decisions [14].

The results have revealed sociomotor regularities in the Hunters' motor behaviour in relation to the Bear. In the following section, we will delve deeper into their strategies by analysing the relationship between socio-affective status and their motor decision-making, allowing for a more precise interpretation of the meaning behind these motor behaviours.

 

## Effect of socio-affective status on the Hunters' motor decisions

The study of the impact of negative socio-affective relationships in the traditional sporting game (TSG) Bear, Guardian and Hunters allows for an analysis of sociometric rejection indices and their connection to the players' motor strategies. In such games, motor actions are not merely physical responses to rules; rather, they are deeply linked to the positions and social relationships between participants [25].

From a sociometric perspective, TSGs operate as social laboratories where dynamics of power, respectful motor interaction, and potential conflicts emerge, reflecting everyday interactions [8]. Within this framework, Kemper's theory of power and status [19] is key to understanding how recognition and autonomy in relationships influence motor strategies, revealing the tension between power and vulnerability across the various roles within the game.

**The Guard distance: a deliberate choice of indifference.** The Bear's socio-affective rejection significantly influenced the distance maintained by the Hunters, highlighting how socio-affective status conditions motor decisions in the game. Statistical analysis shows that in the presence of high rejection (BRejHigh), Hunters opted for medium or long distances, avoiding close approach. Short-distance displacements decreased (43.6% compared to 47.9% in low rejection), while long-distance motor displacements increased (13.6% versus 7.8%), reflecting a strategy of distancing.

This behaviour may be interpreted through the principle of "I've rejected him, therefore I don't play with him", leading Hunters to avoid close interaction with the Bear, despite the fact that the game's objective involves approaching the Bear's intimate social space [8] in order to strike. This dynamic mirrors situations described in other studies on Pelota Sentada (Sitting Ball) [8,46], an unstable and fluctuating ambivalent communication network game, where rejected players are ignored, generating a kind of social indifference. Some results challenge usual expectations regarding the participation of rejected players. Specifically, analysis of motor interactions revealed that certain players rejected in the sociometric questionnaire (indicating that they were not popular or well accepted within the group) received very few interactions during the game.

Moreover, players with high popularity in the questionnaire also experienced antagonistic interactions during the game, such as receiving more direct throws, indicating that social status outside the game does not always translate directly into in-game behaviour. In our study, rejection of the Bear did not translate into increased motor aggressiveness, but rather into a strategy of distancing that preserves the Bear's vulnerable role and exclusion within the group [16,17].

The vulnerability of the Bear, both in their role within the game and in their socio-affective status, leads the Hunters to adopt behaviours that do not aim for direct attack, but rather for maintaining a distance that avoids confrontation. This motor behaviour reflects how socio-affective rejection does not always imply an aggressive response; at times, it manifests through avoidance, thereby conditioning the interactions and strategies within the game. This observation aligns with Goffman's theory of social interaction [47], whereby actors tend to regulate proximity and hostility according to group norms and social status.

The game Bear, Guardian and Hunters exemplifies a complex motor interaction in which affectivity and motricity define the power of those who dominate and the vulnerability of those who are subjected. Understanding motor conduct requires a multidimensional perspective that integrates socio-affective factors and motor strategies [11]. The results show that, unlike the Bear, social rejection does not condition the motor conduct of the Hunters, who retain the autonomy to either approach or distance themselves. This freedom of action confirms their dominant role in the game's dynamic. According to Kemper [19], power is expressed in the ability to exert control during interaction, whereas the Bear's weakened status reinforces the Hunters' distancing strategy, thereby consolidating the game's hierarchy.

## The blows delivered by the Hunters: An asserted power

Socio-affective rejection significantly influences the type of hitting performed by the Hunters. Those with high rejection levels strike more frequently and with greater intensity, showing statistically significant differences. Specifically, 6.8% of the rejected Hunters executed strong hits, compared to only 1.3% of the less rejected ones, suggesting that an excluded status encourages greater motor aggressiveness.

This pattern supports Kemper's theory [19], which links low status with aggressive responses as a mechanism for compensation or reaffirmation of control. This ritualised aggressiveness, also observed by Bredemeier et al. [48], indicates that excluded Hunters seek to consolidate their presence through more intense motor decisions, adapting to the socio-affective roles within the game.

The Bear's rejection did not significantly influence the type of hit received. Regardless of the Bear's socio-affective status, Hunters maintained similar frequencies and intensities of hitting, reinforcing the Bear's constant vulnerability and demonstrating that their rejection does not alter the Hunters' motor aggressiveness.

In this context, the Bear functions as an outlet for ritualised aggression — a form of controlled motor opposition that allows players to express competitiveness without transgressing the rules [14]. The socio-affective dimension influenced offensive motor decisions: in the vulnerable role (Bear), it accentuated their vulnerability, while in the dominant roles (Hunters), it reaffirmed their power.

The multidimensional analysis reveals how the game's structure reinforces the dominance of the Hunters and the passivity of the Bear, consolidating hierarchies where affectivity and motricity are interconnected. Socio-affective roles and status not only influence individual actions, but also shape a dynamic in which motor decisions reflect the tension between power and vulnerability in a context of competition and exclusion. According to Kemper [19], the relationship between power and status in social interaction generates dynamics of inclusion and exclusion: the Hunters reinforce their dominant position by maintaining their freedom of action, while the Bear is relegated to a role of lesser influence and acceptance. This dynamic aligns with Elias and Dunning [15], who highlight that game structures and their shared rules allow for the controlled expression of dominance without transgressing respect.

*Between power and vulnerability: the sociomotor role at play.* Socio-affective status influences the motor decisions of the Hunters, determining both their distance from the Bear and the type of hitting employed. The Bear's vulnerability — both within the game and in the social dimension — leads Hunters to avoid approaching when the Bear is rejected, aligning with studies on violence and exclusion in sporting contexts [10,49]. The analysis of hitting confirms that the Bear's social status does not affect the Hunters' decisions, as they continue to consider the Bear their preferred target for asserting their power. Regardless of the Bear's socio-affective rejection, the Hunters maintain their authority and choose to engage in the game, reinforcing their status within the game's dynamic.

*The role of the Bear: vulnerability in the face of controlled aggression.* As a passive and vulnerable figure, the Bear becomes the primary target of the Hunters, who channel their motor aggressiveness in a controlled manner. This ritualised aggression, described by Parlebas [14], allows for the expression of competitiveness without breaching the game's rules. In this context, motor opposition acquires a meaning beyond physical execution, fitting into Goffman's [21] notion of ritualised motor aggression. Furthermore, the Bear's role generates intense emotional experiences, such as fear or frustration, owing to their vulnerability — a key factor in the game's socio-affective dynamic [11].

The role of Hunters: Influence of socio-affective status on motor action. The negative sociometric status of the Hunters — that is, their level of rejection — influences their motor decisions. Less rejected Hunters adopt a more conservative and less aggressive behaviour, while those more rejected take greater risks and employ more competitive and aggressive strategies. This pattern is consistent with studies on aggressiveness in sport and physical education, where lower social status drives more competitive and risk-prone behaviours as a form of compensation or affirmation within the group [48,50].

Although the *Guardian* role was not included in the empirical analysis of this study, it is worth noting its theoretical relevance for understanding the internal logic of the game. Within the framework of Motor Praxiology, the *Guardian* functions as a moderator of motor action and aggressiveness, regulating the behaviour of the *Hunters* and compelling them to assess risk during their motor actions and decisions. As suggested by Dugas (2017), this role contributes to balancing competitive interactions, ensuring that motor aggressiveness is channelled and regulated within the boundaries of the game. Recognising this theoretical dimension complements the interpretation of the results and provides a broader understanding of how the structure of TSGs promotes affective regulation and social harmony among participants.

*The Bear and Guardian game: a system of sociomotor roles and socioaffective decisions.* The Bear and Guardian game distributes sociomotor roles that involve motor conducts in which socio-affective decisions determine the action. Motor behaviours respond to choices and rejections based on the power assumed, according to the internal logic of the game. This system of socio-affective-motor interactions promotes value education, aligning with Elias and Dunning's [15] view on the civilising role of sport in the regulation of aggressiveness and the acceptance of rules. The structure of the game evokes relational feelings that oscillate between power and vulnerability within the boundaries of respect for the rules, contributing to the development of motor intelligence and the ability to adapt to everyday life situations.

## Conclusion

This study analyses the internal logic of the traditional sporting game (TSG) Bear, Guardian and Hunters, exploring the relationship between socio-affective dynamics and motor decisions in terms of time, space, and interaction. The results show that the Hunters adjust their motor strategies according to the phases of the game: they begin with risky behaviour that evolves into more conservative strategies, increasing the distance from the Bear and reducing motor intensity. This transition reveals how the game phases determine the evolution of motor decisions.

From a sociomotor perspective, the roles within the game influence the interactions. The Bear, as a vulnerable figure, and the Guardian, as a moderator of motor aggressiveness, structure the group dynamic, while the Hunters adjust their actions according to the perceived threat of the Guardian's presence, fostering motor self-regulation in line with the sociomotor and socio-affective dynamics.

The sociometric analysis shows that socio-affective rejection influences the motor decisions of the Hunters: those with higher rejection adopt aggressive and risky strategies, while those less rejected display more moderate and conservative behaviours. These findings highlight the importance of sociometry in understanding sociomotor dynamics in TSGs, supporting Kemper's (2006) [19] theories of power and status and Collins' (2009) [22] theory of ritualised interaction, where inclusion and exclusion shape group interaction and behaviour.

Although some effect sizes were small, the results still indicate meaningful associations, particularly between game phases and spatial choices, social rejection and motor aggressiveness, and interpersonal status and proximity to the Bear. These findings underscore the role of socio-affective variables in shaping motor decisions.

## Limits and future research

This study presents five limitations:

a) The sample consisted of 10 university students, which may seem limited. However, each participant was analysed in depth, with 1,810 game sequences coded and examined. This intensive observational approach provides highly detailed and reliable data on individual and group sociomotor dynamics, even with a few participants. Previous research in motor praxeology and observational studies of traditional sporting games supports the use of small samples when detailed coding of sequences is employed [35].

b) The small sample size and its exclusive origin from university students in sports disciplines restrict the generalization of the results to other cultural contexts and non-athletic populations. Furthermore, a non-probabilistic method was used for participant selection, specifically convenience sampling, which could suggest that we are dealing with an exploratory study.

c) The game duration of eight minutes may also represent a limitation, as a longer time could potentially reveal additional nuances of social interaction. However, the choice of this duration was grounded in previous studies that employed similar intervals (7–8 minutes) to analyse social dynamics in recreational and educational contexts

[6,11,34,35,46]. In addition, maintaining an eight-minute period ensured participant engagement and prevented fatigue, allowing for consistency and comparability with prior research conducted by the Motor Action Research Group (GIAM) [6,29,34,35].

d)  The role of emotions in motor decision-making was not explored in depth, which would have allowed for a more comprehensive understanding of the impact of interactions within the game.

e)  The gender perspective, still under analysis, could have facilitated an examination of how motor and sociometric dynamics vary between male and female players.

For future research, it is recommended to expand the sample and its diversity, including participants from non-athletic backgrounds, as well as to analyse the relationship between emotions and motor decisions, evaluating their impact on behaviour and group cohesion.

This approach will contribute to a deeper understanding of the socialisation processes and the factors shaping motor behaviour in TSGs, offering valuable tools for educators and researchers in motor praxeology.

**Transfer for physical education teachers**

The study of the TSG Bear, Guardian and Hunters offers pedagogical implications for physical education, supporting the development of socio-affective competencies, the regulation of motor aggressiveness, and group cohesion. Despite being conducted with a small, experienced, and sport-oriented sample, this research should be considered as a valuable reference and expanded in future studies to examine its effects in educational contexts.

It is recommended to use games involving role exchange, allowing students to experience different perspectives: the vulnerability of the Bear, the channelling of aggressiveness by the Guardian, and the power of the Hunter. The alternation of roles can serve as a pedagogical tool to help students better understand group dynamics, recognise both collective and individual challenges, and foster interpersonal relationships and social inclusion, consistent with the internal logic of the game.

TSGs enable teachers to analyse socio-affective relationships in the classroom. Rejected students tend to adopt riskier strategies, while those less rejected opt for more conservative behaviours. These games promote inclusion and group cohesion, integrating moments of reflection on each student's role.

Regarding the management of motor aggressiveness, TSGs channel energy within a safe and regulated framework. Teachers can use contact games with clear rules, encouraging a culture of fair play and promoting emotional reflection after the activity.

The study reveals that rejected students actively participate in TSGs as a means of integration, providing an opportunity to foster inclusion. Activities with role exchange enhance their value and sense of belonging to the group. In conclusion, Bear, Guardian and Hunters is an educational tool that develops motor and socio-affective competences, promoting an inclusive environment and respectful coexistence—essential for the holistic development of students.

## Supporting information

**S1 File. Inclusivity-in-global-research-questionnaire.**
(DOCX)

## Acknowledgments

This work was supported of the National Institute of Physical Education of Catalonia (INEFC) of the Generalitat de Catalunya. The authors would like to thank the members of the Research Group on Motor Action (GIAM) of the INEFC.

## Author contributions

**Conceptualization:** Zhaïra Ben Chaâbane, Carlos Mallén-Lacambra, Pere Lavega-Burgués.

**Data curation:** Carlos Mallén-Lacambra, Aaron Rillo-Albert, Cristòfol Salas-Santandreu, Felipe Menezes-Fagundes.

**Formal analysis:** Aaron Rillo-Albert, Cristòfol Salas-Santandreu, Felipe Menezes-Fagundes, Verónica Alcaraz-Muñoz, Miguel Pic, Rosa Rodríguez-Arregi.

**Funding acquisition:** Pere Lavega-Burgués.

**Investigation:** Carlos Mallén-Lacambra, Aaron Rillo-Albert, Pere Lavega-Burgués.

**Methodology:** Zhaïra Ben Chaâbane, Cristòfol Salas-Santandreu, Pere Lavega-Burgués.

**Project administration:** Zhaïra Ben Chaâbane, Carlos Mallén-Lacambra, Pere Lavega-Burgués.

**Supervision:** Zhaïra Ben Chaâbane, Aaron Rillo-Albert, Miguel Pic, Pere Lavega-Burgués.

**Validation:** Carlos Mallén-Lacambra, Aaron Rillo-Albert, Cristòfol Salas-Santandreu, Felipe Menezes-Fagundes, Verónica Alcaraz-Muñoz, Miguel Pic, Rosa Rodríguez-Arregi.

**Visualization:** Verónica Alcaraz-Muñoz, Rosa Rodríguez-Arregi.

**Writing – original draft:** Zhaïra Ben Chaâbane, Carlos Mallén-Lacambra, Aaron Rillo-Albert, Cristòfol Salas-Santandreu, Pere Lavega-Burgués.

**Writing – review & editing:** Cristòfol Salas-Santandreu, Felipe Menezes-Fagundes, Verónica Alcaraz-Muñoz, Miguel Pic.

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
