## [Decision Letter · Decision Letter 0]

23 Sep 2025

Dear Dr. Lavega-Burgues,

Thank you for submitting your manuscript to PLOS ONE. After careful consideration, we feel that it has merit but does not fully meet PLOS ONE’s publication criteria as it currently stands. Therefore, we invite you to submit a revised version of the manuscript that addresses the points raised during the review process.

We look forward to receiving your revised manuscript.

Kind regards,

Hesam Ramezanzade, Ph.D

Academic Editor

PLOS ONE

**Journal Requirements:**

1. When submitting your revision, we need you to address these additional requirements. Please ensure that your manuscript meets PLOS ONE's style requirements, including those for file naming. The PLOS ONE style templates can be found at https://journals.plos.org/plosone/s/file?id=wjVg/PLOSOne_formatting_sample_main_body.pdf and https://journals.plos.org/plosone/s/file?id=ba62/PLOSOne_formatting_sample_title_authors_affiliations.pdf 2. Please include a complete copy of PLOS’ questionnaire on inclusivity in global research in your revised manuscript. Our policy for research in this area aims to improve transparency in the reporting of research performed outside of researchers’ own country or community. The policy applies to researchers who have travelled to a different country to conduct research, research with Indigenous populations or their lands, and research on cultural artefacts. The questionnaire can also be requested at the journal’s discretion for any other submissions, even if these conditions are not met.  Please find more information on the policy and a link to download a blank copy of the questionnaire here: https://journals.plos.org/plosone/s/best-practices-in-research-reporting. Please upload a completed version of your questionnaire as Supporting Information when you resubmit your manuscript. 3. We note that the grant information you provided in the ‘Funding Information’ and ‘Financial Disclosure’ sections do not match.  When you resubmit, please ensure that you provide the correct grant numbers for the awards you received for your study in the ‘Funding Information’ section. 4. Thank you for stating the following financial disclosure: This research was funded by the National Institute of Physical Education of Catalonia (INEFC; code: DOGC N. 8568–22.12.2021Resolution 21_03_2022), Universitat de Lleida (UdL), through the project “OPPORTUNITY. Fostering social inclusion and gender equality in formal and nonformal educational contexts through applying traditional sports and games”, co-funded by the Erasmus+ Programme of the European Union, project code: 622100-EPP-1-2020-1-ES-SPO-SCP.   Please state what role the funders took in the study.  If the funders had no role, please state: "The funders had no role in study design, data collection and analysis, decision to publish, or preparation of the manuscript." If this statement is not correct you must amend it as needed. Please include this amended Role of Funder statement in your cover letter; we will change the online submission form on your behalf. 5. Thank you for stating the following in the Acknowledgments Section of your manuscript: This work was supported by the Institut Nacional d’Educació Física de Catalunya (INEFC) of the Generalitat de Catalunya (Catalonia, Spain). The authors would like to thank the members of the Research Group on Motor Action (GIAM) of the INEFC. We note that you have provided funding information that is not currently declared in your Funding Statement. However, funding information should not appear in the Acknowledgments section or other areas of your manuscript. We will only publish funding information present in the Funding Statement section of the online submission form. Please remove any funding-related text from the manuscript and let us know how you would like to update your Funding Statement. Currently, your Funding Statement reads as follows: This research was funded by the National Institute of Physical Education of Catalonia (INEFC; code: DOGC N. 8568–22.12.2021Resolution 21_03_2022), Universitat de Lleida (UdL), through the project “OPPORTUNITY. Fostering social inclusion and gender equality in formal and nonformal educational contexts through applying traditional sports and games”, co-funded by the Erasmus+ Programme of the European Union, project code: 622100-EPP-1-2020-1-ES-SPO-SCP. Please include your amended statements within your cover letter; we will change the online submission form on your behalf. 6. We note that your Data Availability Statement is currently as follows: All relevant data are within the manuscript and its Supporting Information files. Please confirm at this time whether or not your submission contains all raw data required to replicate the results of your study. Authors must share the “minimal data set” for their submission. PLOS defines the minimal data set to consist of the data required to replicate all study findings reported in the article, as well as related metadata and methods (https://journals.plos.org/plosone/s/data-availability#loc-minimal-data-set-definition). For example, authors should submit the following data: - The values behind the means, standard deviations and other measures reported;- The values used to build graphs;- The points extracted from images for analysis. Authors do not need to submit their entire data set if only a portion of the data was used in the reported study. If your submission does not contain these data, please either upload them as Supporting Information files or deposit them to a stable, public repository and provide us with the relevant URLs, DOIs, or accession numbers. For a list of recommended repositories, please see https://journals.plos.org/plosone/s/recommended-repositories. If there are ethical or legal restrictions on sharing a de-identified data set, please explain them in detail (e.g., data contain potentially sensitive information, data are owned by a third-party organization, etc.) and who has imposed them (e.g., an ethics committee). Please also provide contact information for a data access committee, ethics committee, or other institutional body to which data requests may be sent. If data are owned by a third party, please indicate how others may request data access. 7. Please amend either the abstract on the online submission form (via Edit Submission) or the abstract in the manuscript so that they are identical. 8. Your ethics statement should only appear in the Methods section of your manuscript. If your ethics statement is written in any section besides the Methods, please delete it from any other section. 9. If the reviewer comments include a recommendation to cite specific previously published works, please review and evaluate these publications to determine whether they are relevant and should be cited. There is no requirement to cite these works unless the editor has indicated otherwise. 

Reviewers' comments:

**Comments to the Author**

1. Is the manuscript technically sound, and do the data support the conclusions?

Reviewer #1: Yes

Reviewer #2: Yes

2. Has the statistical analysis been performed appropriately and rigorously?

Reviewer #1: Yes

Reviewer #2: Yes

3. Have the authors made all data underlying the findings in their manuscript fully available?

Reviewer #1: Yes

Reviewer #2: Yes

4. Is the manuscript presented in an intelligible fashion and written in standard English?

Reviewer #1: Yes

Reviewer #2: Yes

**Reviewer #1: ** While the overall quality of the paper is high, addressing a few minor points could elevate its impact and comprehensiveness to an outstanding level:

Sample Specificity and Generalizability: The participant pool, drawn exclusively from athletic disciplines, is a very specific population. This rightly raises the question of whether the same patterns would be observed in a non-athletic or general population. I strongly recommend explicitly acknowledging this as a key limitation for generalizability. Furthermore, it would be excellent to suggest this specific point—investigating these effects in non-athletes—as a direction for future research in your conclusion.

Clarification on the Rejection Calculation: The use of a 'closeness index' is an interesting approach. To further strengthen the Methods section, a small but important clarification would be helpful: please specify the exact threshold used to categorize participants into 'High Rejection' vs. 'Low Rejection' groups. Was this based on a statistical measure like the median, or was a pre-defined theoretical cut-off point used? Adding this single sentence would provide valuable clarity and enhance the replicability of your excellent methodology.

Avoiding Repetition: Some instances of result repetition between the text and tables were noted. To enhance conciseness, you could direct the reader to the relevant table (e.g., "As presented in Table 3...") and focus the text on providing a broader interpretation and analysis of the results, rather than re-stating the statistics.

Nuancing the Interpretation of Statistical vs. Practical Significance: The manuscript occasionally emphasizes statistical significance (the *p*-value) without an equal focus on the practical importance of the finding. For instance, some results, while statistically significant, show a rather weak effect size (e.g., Cramér’s V = 0.054). It would strengthen the discussion to interpret such findings with a bit more caution. Acknowledging that while a result is statistically reliable, its practical or real-world impact within the studied context may be limited would provide a more nuanced and accurate interpretation. This doesn't diminish the finding but rather frames it with appropriate scientific prudence.

**Reviewer #2:**  Dear Editor

This paper addresses an intriguing and relatively novel topic that combines the analysis of motor behavior in traditional games with socio-affective dynamics. The scientific quality of the article is generally acceptable, featuring a rigorous methodological design and appropriate statistical analysis tools. However, the paper has limitations regarding sample size and generalizability of results. A robust theoretical framework has been established based on Parlebas' motor praxeology theory, though some claims require additional empirical support.

Introduction

• Some claims (such as the role of traditional games in interpersonal relationship education) are presented without sufficient empirical evidence.

• Limited literature review with insufficient examination of previous similar research.

Methodology

• Small sample size (10 participants), which the authors themselves acknowledge as a limitation. Is there a convincing explanation or reliable source for this?

• Game duration (8 minutes) may be insufficient for complete understanding of social dynamics. On what basis was this time period chosen?

• Lack of control for confounding variables such as participants' prior game experience.

Results

• Some effect sizes are small (Cramér's V < 0.1), which may limit practical significance. Your interpretation is very important when the effect size is small, which is less addressed.

• No statistical power analysis provided for interpreting non-significant results.

• Lack of examination of more complex interaction effects between variables.

Discussion

• Some interpretations extend beyond the available data. The discussion should be based on the results of this research. The results can be compared with other research, but our criterion for discussion is the results of our research.

• Insufficient discussion of study limitations. It is recommended that research recommendations be presented with an emphasis on the limitations of the research.

• Limited comparison with similar research in other domains. Please strengthen this issue in the discussion.

**Do you want your identity to be public for this peer review?** For information about this choice, including consent withdrawal, please see our Privacy Policy

Reviewer #1: No

Reviewer #2: No

---

## [Author Response · Author response to Decision Letter 1]

22 Oct 2025

Dear Editor and Reviewers,

We highly appreciate your valuable feedback. We consider your comments to be pertinent, and we would like to sincerely thank you for the effort and careful evaluation of our manuscript.

Please find attached our detailed response to the editor and reviewers, prepared in accordance with the Journal Requirements and the Review Comments to the Author. In this document, we explain how we have addressed each of the suggestions and requirements, which have also been incorporated into the revised manuscript and the online submission form.

Journal Requirements:

We meet the style requirements.

We believe that the inclusivity questionnaire does not apply to our study, as it was conducted entirely within our own academic community.

We corrected the mistake.

This research was funded by the National Institute of Physical Education of Catalonia (INEFC; code: DOGC N. 8568–22.12.2021Resolution 21_03_2022), Universitat de Lleida (UdL), through the project “OPPORTUNITY. Fostering social inclusion and gender equality in formal and nonformal educational contexts through applying traditional sports and games”, co-funded by the Erasmus+ Programme of the European Union, project code: 622100-EPP-1-2020-1-ES-SPO-SCP.

We included the recommended sentence on the funding statement: "The funders had no role in study design, data collection and analysis, decision to publish, or preparation of the manuscript."

This work was supported by the Institut Nacional d’Educació Física de Catalunya (INEFC) of the Generalitat de Catalunya (Catalonia, Spain). The authors would like to thank the members of the Research Group on Motor Action (GIAM) of the INEFC.

This research was funded by the National Institute of Physical Education of Catalonia (INEFC; code: DOGC N. 8568–22.12.2021Resolution 21_03_2022), Universitat de Lleida (UdL), through the project “OPPORTUNITY. Fostering social inclusion and gender equality in formal and nonformal educational contexts through applying traditional sports and games”, co-funded by the Erasmus+ Programme of the European Union, project code: 622100-EPP-1-2020-1-ES-SPO-SCP.

We understand your message, but our university's policy on funding scientific publications states the following: All published articles must have the following phrase in the acknowledgements: "With the support of the National Institute of Physical Education of Catalonia (INEFC) of the Generalitat de Catalunya". Is it possible to keep the sentence in this section so as not to have problems justifying the publication of the article in our university? If possible, we would like to keep that phrase. In this case, the University provides us with a small grant to help cover the costs of publishing the article, which has nothing to do with funding the design and implementation of the research.

6. We note that your Data Availability Statement is currently as follows: All relevant data are within the manuscript and its Supporting Information files.

Along with the comments, we will send an Excel document with the data.

7. Please amend either the abstract on the online submission form (via Edit Submission) or the abstract in the manuscript so that they are identical.

Attention will be paid to this consideration.

8. Your ethics statement should only appear in the Methods section of your manuscript. If your ethics statement is written in any section besides the Methods, please delete it from any other section.

We removed ethical information from statements and declarations section.

Reviewers did not include this kind of recommendations.

10. Review Comments to the Author

Reviewer #1: While the overall quality of the paper is high, addressing a few minor points could elevate its impact and comprehensiveness to an outstanding level:

Sample Specificity and Generalizability: The participant pool, drawn exclusively from athletic disciplines, is a very specific population. This rightly raises the question of whether the same patterns would be observed in a non-athletic or general population. I strongly recommend explicitly acknowledging this as a key limitation for generalizability. Furthermore, it would be excellent to suggest this specific point—investigating these effects in non-athletes—as a direction for future research in your conclusion.

We greatly appreciate your comment. We fully agree that the specificity of the sample constitutes a limitation for the generalization of the findings. Consequently, we have incorporated the following text in the section on “Limits and future research”:

a) The small sample size and its exclusive origin from university students in sports disciplines restrict the generalization of the results to other cultural contexts and non-athletic populations.

For future research, it is recommended to expand the sample and its diversity, including participants from non-athletic backgrounds, as well as to analyse the relationship between emotions and motor decisions, evaluating their impact on behaviour and group cohesion.

Clarification on the Rejection Calculation: The use of a 'closeness index' is an interesting approach. To further strengthen the Methods section, a small but important clarification would be helpful: please specify the exact threshold used to categorize participants into 'High Rejection' vs. 'Low Rejection' groups. Was this based on a statistical measure like the median, or was a pre-defined theoretical cut-off point used? Adding this single sentence would provide valuable clarity and enhance the replicability of your excellent methodology.

We greatly appreciate this observation. We have incorporated the following clarification in the “Methods” section:

“The classification was made using the midpoint of the empirical range of the observed scores (minimum = 9, maximum = 45). Participants with a score below 27 on proximity to rejection were classified in the High Rejection group, while those with a score above 27 were placed in the Low Rejection group. It should be noted that proximity to rejection is an inverse indicator, so lower values reflect higher levels of rejection.”

Avoiding Repetition: Some instances of result repetition between the text and tables were noted. To enhance conciseness, you could direct the reader to the relevant table (e.g., "As presented in Table 3...") and focus the text on providing a broader interpretation and analysis of the results, rather than re-stating the statistics.

Nuancing the Interpretation of Statistical vs. Practical Significance: The manuscript occasionally emphasizes statistical significance (the *p*-value) without an equal focus on the practical importance of the finding. For instance, some results, while statistically significant, show a rather weak effect size (e.g., Cramér’s V = 0.054). It would strengthen the discussion to interpret such findings with a bit more caution. Acknowledging that while a result is statistically reliable, its practical or real-world impact within the studied context may be limited would provide a more nuanced and accurate interpretation. This doesn't diminish the finding but rather frames it with appropriate scientific prudence.

Thank you very much, dear reviewer. We have nuanced the statistical relevance more appropriately. You will find these nuances in various places throughout the document. Specifically, in: (i) the results section, where the more descriptive analysis concluded with a practical comment, in line with the reference theoretical framework, and also in (ii) the discussion section

Reviewer #2: Dear Editor

This paper addresses an intriguing and relatively novel topic that combines the analysis of motor behaviour in traditional games with socio-affective dynamics. The scientific quality of the article is generally acceptable, featuring a rigorous methodological design and appropriate statistical analysis tools. However, the paper has limitations regarding sample size and generalizability of results. A robust theoretical framework has been established based on Parlebas' motor praxeology theory, though some claims require additional empirical support.

Introduction

• Some claims (such as the role of traditional games in interpersonal relationship education) are presented without sufficient empirical evidence.

We greatly appreciate your comment. We added the following sentence:

“Previous research has shown that educational interventions with TSGs can enhance group dynamics and strengthen interpersonal relationships among students (6). At the same time, other studies emphasize that pre-existing social configurations shape the motor interactions that take place during the game (7). Together, these findings suggest that TSGs operate as a genuine relational laboratory, providing meaningful opportunities to foster students’ education in peaceful coexistence.”

• Limited literature review with insufficient examination of previous similar research.

We greatly appreciate your comment.

We added empirical evidence in this paragraph: “This analysis shows that motor strategies depend on socio-affective status (6), providing an internal insight into the meaning of ludic action.”

And complemented another paragraph with the following information: “In this line, previous studies on this game reported significant differences in the way men and women executed hitting actions, influenced by prevailing gender norms (6).”

Methodology

• Small sample size (10 participants), which the authors themselves acknowledge as a limitation. Is there a convincing explanation or reliable source for this?

We appreciate your comment. The sample for this study consists of 10 students due to the detailed observational approach applied. Each participant was analysed in depth, recording 1,810 game sequences, which required many hours of coding work. This intensive design allows for very rich and reliable data on individual and group sociomotor dynamics, even with a small number of participants. Previous studies in motor praxeology and observational analysis of traditional sports games have shown that, when detailed observation sequences are used, small samples can provide robust and meaningful findings (35). https://doi.org/10.3389/fpsyg.2020.01384

Other examples of rigorous research that employed an observational methodology also used a reduced number of participants. Follow-up on 6 adolescents (Arias-Pujol and Anguera, 2017), 14 football players (Casamichana, Castellano et al., 2012), or 2 sets of a tennis match, using 1100 observation records (Plaza, Hernández-Mendo et al, 2006) are examples. It can be understood that the observational methodology demands a significant amount of time and the use of a series of resources that in some cases go beyond the researchers' sustainability

Arias-Pujol E and Anguera MT (2017) Observation of Interactions in Adolescent Group Therapy: A Mixed Methods Study. Front. Psychol. 8:1188. doi: 10.3389/fpsyg.2017.01188

Casamichana, D., Castellano, J., Blanco-Villaseñor, Á., & Usabiaga, O. (2012). Estudio de la percepción subjetiva del esfuerzo en tareas de entrenamiento en fútbol a través de la teoría de la generalizabilidad. Revista de psicología del deporte, 21(1), 0035-40.

Plaza, J. O. G., Hernández-Mendo, A., & Morales-Sánchez, V. (2006). Sistema de codificación y análisis de la calidad del dato en el tenis de dobles. Revista de psicología del deporte, 15(2), 279-294.

We have added another paragraph to the limitations section that addresses this issue:

The sample consisted of 10 university students, which may seem limited. However, each participa

---

## [Editor Report · Decision Letter 1]

3 Nov 2025

The Sociomotor Role and its Meaning in Traditional Sporting Games: Socio-affective Rejection in Question

PONE-D-25-33331R1

Dear Dr. Lavega-Burgues,

We’re pleased to inform you that your manuscript has been judged scientifically suitable for publication and will be formally accepted for publication once it meets all outstanding technical requirements.

Kind regards,

Hesam Ramezanzade, Ph.D

Academic Editor

PLOS ONE
---

## [Editor Report · Acceptance letter]

PONE-D-25-33331R1

PLOS ONE

Dear Dr. Lavega-Burgués,

I'm pleased to inform you that your manuscript has been deemed suitable for publication in PLOS ONE. Congratulations! Your manuscript is now being handed over to our production team.

Kind regards,

on behalf of

Dr. Hesam Ramezanzade

Academic Editor

PLOS ONE